# SARS-CoV-2 Infection: A Clinical and Histopathological Study in Pregnancy

**DOI:** 10.3390/biology12020174

**Published:** 2023-01-21

**Authors:** Angelica Perna, Eleonora Hay, Paolo De Blasiis, Marco La Verde, Francesca Caprio, Marco Torella, Maddalena Morlando, Carmine Sellitto, Germano Guerra, Angela Lucariello, Alfonso Baldi, Antonio De Luca

**Affiliations:** 1Department of Medicine and Health Sciences “Vincenzo Tiberio”, University of Molise, 86100 Campobasso, Italy; 2Department of Mental and Physical Health and Preventive Medicine, Section of Human Anatomy, University of Campania “Luigi Vanvitelli”, 80138 Naples, Italy; 3Obstetrics and Gynecology Unit, Department of Woman, Child and General and Specialized Surgery, University of Campania “Luigi Vanvitelli”, 80138 Naples, Italy; 4Department of Sport Sciences and Wellness, University of Naples “Parthenope”, 80133 Naples, Italy; 5Department of Environmental, Biological and Pharmaceutical Sciences and Technologies, Università degli Studi della Campania “L. Vanvitelli”, 81100 Caserta, Italy

**Keywords:** COVID-19, pregnancy, placenta, immune cells, placenta vascular abnormalities, placental histological alterations

## Abstract

**Simple Summary:**

SARS-CoV-2 infection is now known to be associated with several adverse events. However, not much is known about the effect it may have if contracted during pregnancy, on the proper development of the placenta or, subsequently, of the fetus itself. Studies have shown that there may be an increased incidence of developing pathological conditions, such as preterm delivery, fetal growth retardation, the onset of gestational diabetes or hypertensive disorders. In this study, several cases of women who became infected at different times during pregnancy (during the first, second or third trimester) were examined, and in particular it was assessed whether the infection had a role in altering the proper vascular development of the placenta. The greatest alterations were observed when infection occurred during the third trimester; alterations in the expression levels of certain markers of vasculogenesis such as the presence of fibrin deposits, lymphocyte infiltration in the villi, edema and thrombi were observed. Further studies are needed, however, as the mechanisms involved are not yet clear and therefore it is not yet possible to think of therapeutic strategies.

**Abstract:**

During pregnancy, SARS-CoV-2 infection is associated with several adverse outcomes, including an increased risk of pre-eclampsia, preterm delivery, hypertensive disorders, gestational diabetes, and fetal growth restriction related to the development of placenta vascular abnormalities. We analyzed human placenta from full-term, uncomplicated pregnancies with SARS-CoV-2 infection during the first, second, or third trimesters of gestation. We studied, by the immunohistochemistry technique, the expression of CD34 and podoplanin (PDPN) as markers of vasculogenesis to find any differences. As secondary outcomes, we correlated maternal symptoms with placental histological alterations, including fibrin deposits, lymphocyte infiltration in the villi, edema, and thrombi. Our results showed a PDPN expression around the villous stroma as a plexiform network around the villous nucleus of fetal vessels; significant down-regulation was observed in the villous stroma of women infected during the third trimester. CD34 showed no changes in expression levels. During SARS-CoV-2 infection, the most common maternal symptoms were fever, anosmia, ageusia and asthenia, and the majority were treated with paracetamol, corticosteroids and azithromycin. Patients that required multiple symptomatic treatments evidenced a large amount of fibrin deposition in the villi. Certainly, PDPN plays a key role in healthy placental vasculogenesis and thus in its proper physiology, and SARS-CoV-2 surely alters its normal expression. Further studies are necessary to understand what mechanisms are being altered to try to avoid possible complications for both the mother and fetus in terms of the contagions that will still occur.

## 1. Introduction

At the end of December 2019, a new form of coronavirus rapidly spread from China around the world, generating what was declared a global pandemic in just a few months [1]. The nomenclature of this new form of coronavirus, SARS-CoV-2 (severe acute respiratory syndrome coronavirus 2) suggests restricted effects on the respiratory organs. Nevertheless, it has been shown that the virus can also cause consequences for numerous non-respiratory organs, such as kidney [2], liver [3], heart [4], male reproductive system [5]. Transmission of pathogens can occur during pregnancy, labor and birth, as well as after birth, through breastfeeding or simple contact with the mother or others. When a new infectious agent is found, the first question is precisely the ability of this to be transmitted through the placenta, causing adverse effects, and the impact it may have on the normal physiology of the organ itself [6]. With regard to the new SARS-CoV-2 infection, transmission appears to be rare [7]. Indeed, for transmission to occur, the pathogen must reach and cross the placenta and the placenta does not appear to co-express high levels of the primary factors that facilitate virus entry, such as angiotensin-converting enzyme 2 (ACE2) and trans-membrane serine protease 2 (TMPRSS2) [8,9].

What is clear from recent systematic reviews and meta-analyses is that SARS-CoV-2 infection during pregnancy is associated with a number of adverse pregnancy outcomes, including a pro-inflammatory state, resulting in systemic endothelial dysfunction [10], that often leads to the development of pre-eclampsia [11,12], hypertensive disorders from preterm delivery, gestational diabetes and low birth weight [13,14,15,16,17,18,19], as well as, dysfunction of the renin-angiotensin system and onset of vasoconstriction through binding to ACE2 receptors [20,21,22,23,24].

Women with severe SARS-CoV-2 disease showed altered serum values in the assay of biomarkers such as fms-like tyrosine kinase (sFlt-1) and placental growth factor (PlGF) [22]. 

The pathological condition that contributes to fetal growth retardation, preterm birth or fetal death is vascular malperfusion of the fetal placenta, which is often found by analysing the histopathological findings of the placentas of patients with COVID-19 at delivery [25]. Moreover, a study in the Netherlands demonstrated a reduction in the incidence of preterm birth through the use of disease mitigation measures [26]. Podoplanin (PDPN) is a mucin-like transmembrane glycoprotein that is required for the proper development of certain structures, such as the pulmonary alveoli, heart and lymphatic system, to name but a few [27]. PDPN has no functional domain or enzymatic activity of its own. It interacts with molecules present on its cell surface and on neighboring cells to exert its ‘activities‘, and examples are C-type lectin-like receptor-2 (CLEC-2), CD44, or protein kinase A (PKA). On lymphatic endothelial cells and fibroblastic reticular cells, the activation of CLEC-2 by PDPN triggers the activation and aggregation of platelets, a necessary condition for the diversification of blood and lymphatic vessels [28], and for cerebrovascular patterning [29]. After completion of vessel development, PDPN participates in the maintenance of endothelial integrity [28]. Up-regulation is observed on tumor cells or immune cells, fibroblasts and epithelial cells during inflammation [27]. In tumors, PDPN induces migration, invasion, metastasis, proliferation and clonal capacity [27,30], and the mechanisms depend on the binding partners. Maternal uterine blood vessels undergo dramatic vascular remodeling during pregnancy to accommodate the increased utero-placental blood flow, which supports the growth and development of the placenta and fetus [31]. Uterine lymphatic vessels are also involved in this process as they regulate blood vessel remodeling and fluid homeostasis in the uterus during pregnancy; however, no lymphatic vessels have been detected in the placenta, so it remains unclear how interstitial fluid homeostasis is regulated in the placenta [32]. Nevertheless, markers associated with lymphatic vessels are expressed in the placenta, such as VEGF-C, VEGF-D, VEGFR-3/Flt-44 and the lymphatic vessel-specific marker PDPN. PDPN is expressed in the placenta throughout pregnancy and is significantly down-regulated in conditions such as pre-eclampsia, suggesting its important involvement in normal placental development; its absence leads to interstitial imbalances [33]. Immunohistochemical analysis is widely used to study the expression of factors involved in the development of the placenta in various conditions [34], both to study the mechanisms involved in the physiological development and maintenance of this organ and to understand the alterations involved in pathological conditions [35]. In the present research work we studied the expression of CD34 and PDPN as markers of vasculogenesis in pregnant infected by SARS-CoV-2 during the pregnancy. In addition, we correlate the period and treatment of maternal symptoms with placental histological abnormalities to explore a potential correlation.

## 2. Material and Methods

### 2.1. Samples

This prospective cohort study was carried out between April to October 2021. All pregnant were recruited from the Department of Obstetrics and Gynecology at the “Luigi Vanvitelli” University Hospital, Naples, Italy. The patients were enrolled before the labor onset. We included uncomplicated full-term pregnancy according to Chappell’s definition: “a normotensive pregnancy, delivered at >37 weeks, ending in a live-born baby who was not small for gestational age and did not have any other notable pregnancy complications” [36]. To assess fetal well-being, computerised cardiotocography was conducted before the onset of labour [37,38,39,40]. We excluded pregnancies affected by fetal structural or chromosomal malformations, stillbirths, preterm labor, fetal growth restriction, cardiotocography alterations and all the pregnancies complicated by maternal pathology. 

To classify the severity of COVID-19 disease in pregnant women, different scientific societies proposed standardized clinical schemes. We followed the American College of Obstetricians and Gynecologists (ACOG) classification, which assesses the severity/symptoms into mild/moderate or severe based on patient history, physical exam findings, laboratory tests, imaging studies, and patient response to treatment. 

Placental sampling was performed following standard protocols and in a methodical way at numerous regions, such as chorionic plate, chorionic villi, basal plate, amniotic membrane, and umbilical cord [41].

Our Institutional Ethical Review Board authorized the research design (Prot. number 0005381/i), and all participating women submitted written consent after a detailed discussion. A total of 19 samples:
Group 1—Control. Three term placentas from women who were not SARS-CoV-2 infected during pregnancy and had no pathologies.Group 2—Infection during the first trimester of gestation. Seven term placentas from women who became SARS-CoV-2-infected during the first trimester of pregnancy.Group 3—Infection during the second trimester of gestation. Two term placentas from women who became SARS-CoV-2-infected during the second trimester of pregnancy.Group 4—Infection during the third trimester of gestation. Seven term placentas from women who became SARS-CoV-2-infected during the third trimester of pregnancy.

In Table 1 and Table 2 are shown maternal and neonatal clinical details regarding the SARS-CoV-2 infection and treatments. 

The infants tested for SARS-CoV-2 were negative and showed no pathologies.

Placenta specimens were fixed in 10% neutral-buffered formalin solution for 12 h and processed for the preparation of paraffin blocks. The samples were cut into 5 μm-thick sections. Hematoxylin–eosin staining was performed for preliminary histopathological analysis. Classical histopathological parameters for the classification of histopathological abnormalities were used for the analysis of placentas. The histopathological analysis considered vascular alterations in placentas: villous fibrin deposits, villous lymphocyte infiltration, edema and thrombus. Observations are given in Table 2 (+ mild degree; ++ moderate degree; +++ intense degree; − not present).

### 2.2. Immunohistochemistry

Immunohistochemistry was essentially carried out as described previously [42]. Briefly, sections, embedded in paraffin, from each specimen were cut to 5 μm, mounted on glass and dried overnight at 37 °C. All sections were then deparaffinized in xylene, rehydrated through a graded series of alcohol and washed in phosphate-buffered saline (PBS 1X). PBS was used for all subsequent washes and for antiserum dilution. Tissue sections were sequentially quenched in 3% hydrogen peroxide and blocked with PBS-6% non-fat dry milk (Biorad) for 1 h at room temperature. Slides were then incubated at 4 °C overnight with anti-CD34 (Ventana Medical Systems, Oro Valley, AZ, USA—JC70) at a concentration of 0.8 μg/mL, or anti-podoplanin (Ventana Medical Systems, Oro Valley, AZ, USA—D2-40) at a concentration of 0.27 μg/mL, antibodies.

After several washes (3 × 5 min) to remove excess antibody, the slides were incubated with ultraview Universal HRP Multimer secondary antibody at a concentration of 55 μg/mL in a buffer containing protein with ProCin 300 for 1 h. All the slides were then processed by the ABC method (Vector Laboratories, Newark, CA, USA) for 30 min at room temperature. DAB (Vector Laboratories, Newark, CA, USA) was used as the chromogen and hematoxylin was used as the nuclear counterstain. Negative controls for each tissue section were prepared by substituting the primary antiserum with non-immune IgG. For each experiment, all slides were stained in a single batch and thus received equal staining. Immunohistochemical staining intensity was evaluated and ranked: −(absent), +(weak), ++(moderate), or +++(intense).

## 3. Results

Sixteen placentas from patients with SARS-CoV-2 were recruited. Seven pregnant women were positive for SARS-CoV-2 in the first trimester, two in the second trimester and seven in the last trimester of pregnancy. All patients were recruited at the time of birth. All the pregnant women were non-smokers, 15 were Caucasian and 1 Asian. The diagnosis of infection was made by nasopharyngeal swab. Once the sample was obtained, its analysis was carried out by real-time fluorescence-based polymerase chain reaction (RT-PCR).

Unfortunately, when the patients were selected, only two cases were in group 3, according to the inclusion criteria. Despite the small number, we still preferred to include them so as to have a fair distribution. However, we are working on expanding the number of cases. 

### 3.1. Correlation between Maternal Symptoms and Histological Changes in the Placenta

Pregnant women have an increased risk of becoming severely ill due to COVID-19 infection. The Centers for Disease Control and Prevention (CDC) has investigated the effects of COVID-19 in pregnant women and identified several risk factors associated with severe illness or death. These include an advanced maternal age, underlying medical conditions such as diabetes, obesity, hypertension, asthma, and smoking during pregnancy. Therefore, these risk factors were included in the COVID-19 classification of pregnant women. The main symptoms and any treatments carried out, as reported by the patients, are summarized in Table 1. 

The histopathological analysis showed seven women with a large amount of fibrin deposition in the villi, compared with group 1, as well as three from groups 2 and 4 and one from group 3 (Table 2); this condition appeared to be associated with a significant severity of SARS-CoV-2 infection that required multiple treatments, and in particular, three patients were treated with low-molecular-weight heparin (Table 1). Furthermore, in group 2, several patients showed a high villous leucocyte infiltrate, edema and thrombus, compared to group 1 (Table 2); a therapy consisting of a combination of 2–4 drugs was required for these patients (Table 1). When analyzing the Apgar scores of the different groups, no differences were found compared to group 1, just as no fetal abnormalities were found so far (Table 1).

### 3.2. PDPN and CD34 Immunoreactivity

In the placenta, PDPN expression was found in the villous and decidua stroma. In detail, specific staining showed a plexiform network pattern around the villous nucleus of the fetal vessels. The expression of PDPN was significantly reduced in the stroma of the placental villi of mothers infected during the third trimester (group 4) and in the stroma of the decidua of mothers infected during the second trimester (group 3), while a slight decrease in expression was observed in the stroma of the decidua of mothers infected during the third trimester of pregnancy (group 4) (Figure 1 and Table 3). CD34 was used as a blood vessel endothelial marker. It was only expressed in the villous nucleus of the fetal vessel endothelium. No differences in CD34 immunoreactivity were observed among placental tissue samples within any study group (Figure 1 and Table 3).

## 4. Discussion

It is not easy to compare the results observed in the different studies conducted on the analysis of histopathological preparations of term placentas from SARS-CoV-2 patients. In fact, there is not much homogeneity in sample collection, design, inclusion and exclusion criteria, and classification of placental lesions [43,44,45,46,47]. Despite this, it is possible to state that, although to varying degrees, alterations are observed at the placental level that may lead to adverse outcomes for the unborn child [44,45,46]. In our SARS-CoV-2 cohort the etiology leading to the lesions observed in the placentas as a result of vascular changes is unclear. However, we can hypothesize the presence of partial or intermittent obstruction in both umbilical and segmental blood flow, such as occlusion of the chorionic plaque and villous stem vessels, resulting in localized thrombosis, fibrin deposits in the villi, lymphocytic infiltrate and edema. How SARS-CoV-2 may be responsible for the alterations in fetal umbilical blood flow is not known but indirect immune-mediated or virus-mediated damage with development of hypoxemia, resulting in coagulopathy and endotheliopathy may be suspected. An important role in the development of lesions is the duration of infection, the gestational period, and certainly an immature placenta is more vulnerable to certain types of viral-mediated damage than a term placenta [48,49]. 

Villitis develops more in women with a longer interval between infection and birth. The cases observed in our study were not related to a medical history that could explain the cause. In particular, three women infected in the first trimester showed a significant leukocyte infiltrate, compared to that observed in some control samples; it could be hypothesised that, in these cases, exposure to SARS-CoV-2 occurred at the beginning of the first trimester, resulting in the development of villitis due to the virus remaining for weeks. The remaining cases overlap with the controls. The viral infection induces an inflammatory state [50] and provokes a systemic immune response with a consequent increase in cytokines that recall leucocytes from the bloodstream; the condition of villitis could be a direct or indirect consequence of this condition [51]. 

Also in group 2, an increased incidence of thrombus formation and edema is observed, compared to controls. Finally, with regard to fibrin deposits, these were particularly evident, respectively, in 3 samples from group 2, 1 from group 3 and 3 from group 4, in relation to the control samples. It is therefore evident that, in relation to these observations, the greatest presence of abnormalities in blood vessels can be found in the group 2 samples, highlighting the greater impact of infection on the early stages of development of structures. In addition, a correlation has been observed between a stronger symptomatology of patients during infection and a greater presence of histological abnormalities of the placenta; for example, a correlation has been observed between the use of drug therapy and a greater deposition of fibrin in the vessels of the villi, regardless of the period of infection. 

Several studies demonstrated that placentas from pregnant women infected with SARS-CoV-2 had histopathologic indicators of placental hypoperfusion and inflammation [47,52,53,54]. PDPN was used in physiological and pathological tissue samples to study lympho-angiogenesis. The endothelial marker CD34 was used as a complementary marker of blood vessels. A compartmental difference was observed for the distribution of PDPN and CD34 in placentas tested. CD34 is localized on the endothelial cells of the placental vessels and exhibited a strong expression pattern in all the samples analyzed, with no differences between cases and controls in all groups. PDPN is specifically expressed in a reticular-like stroma complex within the exchange and conducting villi. Moreover, it is localized in the stroma of the decidua. When comparing expression levels with controls, a down-regulation is observed, particularly in group 3 significant in the decidua stoma, while in group 4 slight in the decidua stoma and significant in the villus stroma. Group 2, on the other hand, is in line with the controls. The localization of PDPN suggests the presence of a lymphatic-like conductive network with ability to maintain homeostasis, although it remains to be clarified which cells (Hoffbauer cells/macrophages or fibroblasts) produce or secrete PDPN [55]. PDPN plays an important role in various physiological and pathological processes, such as inflammation, thrombus formation and cancer progression. In epithelial and mesenchymal cells, during ischemia-hypoxia, inflammation and cancer, its expression is up-regulated, and relevant agents are growth factors (VEGF-C and TGF-β1), cytokines (TNF-α, IFN-γ, IL-1 and IL6), fibronectin and lipopolysaccharides [30]. It is also known that the lack of PDPN leads to alterations in normal angiogenic development [55]. In addition, podoplanin-secreting cells in the placental tissue, chorionic villus stroma and decidual stromal cells play a role in feto-maternal immunological tolerance [56]. Several studies have shown that PDPN is poorly expressed in tissues in the first few weeks of gestation, in which the fetal vascular network has not yet been established, whereas, as the gestational period progresses, its expression, in the stroma of the chorionic villi, increases [57]. In the decidualized endometrium, immunoreactivity is observed in the cytoplasm and on the cell membrane, most evident around the spiral arterioles. Very rarely, PDPN immunoreactivity was observed in Hofbauer cells and in the apico-luminal region of the syncytiotrophoblast [57]. Evaluation of PDPN expression levels in pathological conditions showed an increase in cases of IUGR; in cases of placental inflammation no change was found, whereas in cases of hydropic placenta there is a reduction in edematous villi and an increase in fibrotic and hydropic villi in the stromal cells around the cysts [57]. In the case of pre-eclampsia there are studies with conflicting results; Wang J. et al. showed that PDPN decreases in the stromal area of villous tissue [55], while Kandemir et al. showed that PDPN increases in the distal villi [57]. Lundell et al. demonstrated that the production of B-cell activation factors by podoplanin-positive decidual stromal cells is important for local B-cell homeostasis during pregnancy [56]. Volchek et al. reported that the expression of PDPN is very strong in decidual cells around the spiral arterioles, an area where lymphatic regression of the gestational endometrium occurs [58]. It is known that PDPN regulates pericyte migration with a role in vascular maturation and function. PDPN plays an important role in interstitial fluid homeostasis. Podoplanin-positive stromal cells in the uveoscleral tissue have been shown to play a role in the formation of lymphatic-like channels for fluid transport [59], while in vivo experiments, in which a mutation in the podoplanin gene was created, resulted in the formation of diffuse lymphoedema [60].

In our study, we found a down regulation of PDPN. This could be interpreted as a blockade of the immunomodulatory function of decidual cells and a less pericyte mobilization resulting in an altered vascular network. All placentas examined are at term; changes in PDPN expression are observed in groups 3 and 4, not in group 2, although vessel changes are more evident in this group. 

Since the first trimester is the most important period for the formation of the placental vasculature, and abnormalities in the vessels are visible in group 2, one might assume an alteration in PDPN expression, as found in groups 3 and 4, and a subsequent compensatory reaction, which is able to bring PDPN levels back to normal, while the abnormalities remain. Probably in groups 3 and 4, being in a period in which the vascular network is already defined, alterations in the vessels are not easily detected while altered PDPN expression persists. 

The reduction of PDPN could be related to a compromised interstitial fluid balance in the placental tissue, resulting in edema formation. Further studies on suitable *in vitro* and *in vivo* models are needed to confirm this hypothesis and the impact of the infection severity on the placenta.

## 5. Conclusions

SARS-CoV-2 infection, contracted during pregnancy, has been shown to have an effect on the development of pathological conditions, such as pre-eclampsia, preterm delivery, hypertensive disorders, gestational diabetes and fetal growth restriction, that are linked to the development of placental vascular abnormalities. This is a preliminary study, as further studies are needed, that demonstrated the presence of possible placental vascular abnormalities resulting from an infection contracted particularly during the third trimester of gestation. A down-regulation of the expression levels of PDPN, a protein involved in the proper development and maintenance of vascular integrity, was observed, while CD34 expression levels were unchanged. The down-regulation of PDPN could be related to a compromised interstitial fluid balance in the placental tissue, resulting in edema formation; the use of other appropriate in vitro and in vivo models could confirm this hypothesis.

## Figures and Tables

**Figure 1 biology-12-00174-f001:**
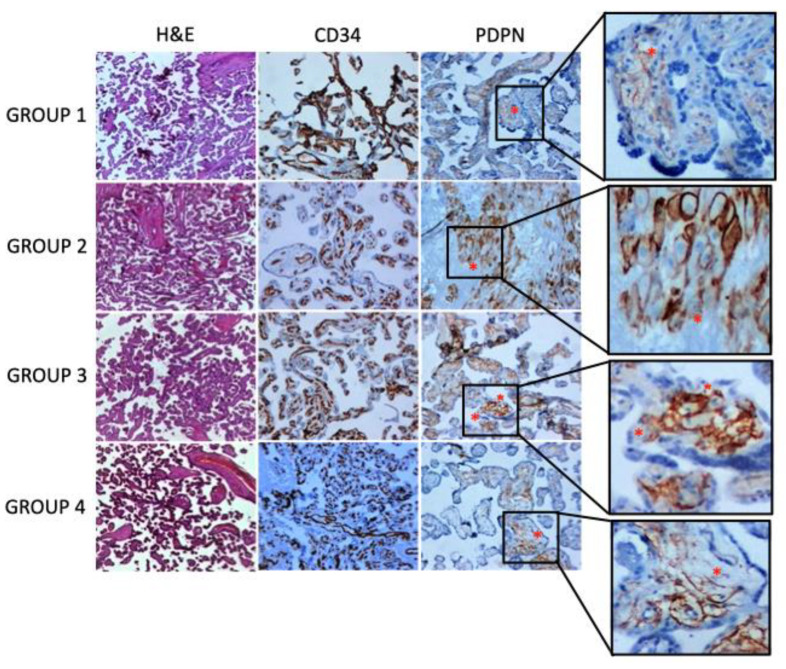
Histology of placental samples at different gestational periods and stained with hematoxylin and eosin is depicted in the first row of the panel. In the second and third rows of the panel, exemplificative immunohistochemical stainings for CD34 and podoplanin are shown. An asterisk indicates areas of podoplanin expression with the characteristic plexiform network pattern around the villous nucleus of the fetal vessels (original magnification ×5).

**Table 1 biology-12-00174-t001:** Clinical details.

	Group 1 (Total Patients = 3)	Group 2 (Total Patients = 7)	Group 3 (Total Patients = 2)	Group 4 (Total Patients = 7)
Maternal age, range (years)	22–31	23–40	29–36	21–40
Gestation at delivery, range (weeks)	38–40	38–40	37–40	37–41
Spontaneous delivery (patient numbers)	2	4	1	4
Caesarean section (patient numbers)	1	3	1	3
Smokers (yes/no)	No	No	No	No
APGAR score 1 min, range APGAR score 5 min, range	8–9 9–10	8–9 9–10	8–10 9–10	8–9 9–10
Birth weight, range (g)	3340–3780	3010–3430	2950–4010	2760–3550
BMI, range	19–21	18–28	19–21	20–23
Duration of infection, range (days)	0	15–21	21–22	18–33
Most common symptoms				
Fever (patient numbers)	0	4	2	2
Rhinitis (patient numbers)	0	1	0	0
Cough (patient numbers)	0	1	1	1
Cephalalgia (patient numbers)	0	1	1	2
Muscle pain (patient numbers)	0	2	1	2
Asthenia (patient numbers)	0	3	1	3
Ageusia/anosmia (patient numbers)	0	1	2	2
Pharyngodynia (patient numbers)	0	0	0	2
Chest pain (patient numbers)	0	1	0	1
Therapy				
Paracetamol (patient numbers)	0	3	3	2
Corticosteroids (patient numbers)	0	3	2	3
Azithromycin (patient numbers)	0	2	2	4
Low-molecular-weight heparin (patient numbers)	0	1	1	1

**Table 2 biology-12-00174-t002:** Histopathological analysis.

	Group 1 (Total Patients = 3)	Group 2 (Total Patients = 7)	Group 3 (Total Patients = 2)	Group 4 (Total Patients = 7)
Villous fibrin deposits (patient numbers)	+ (2) ++ (1)	+ (2) ++ (2) +++ (3)	+ (1) +++ (1)	+ (1) ++ (3) +++ (3)
Villous lymphocyte infiltration (patient numbers)	+ (2) ++ (1)	+ (2) ++ (2) +++ (3)	+ (1) ++ (1)	+ (4) ++ (3)
Villous edema (patient numbers)	+ (3)	+ (3) +++ (4)	+ (1) − (1)	+ (2) ++ (4) − (1)
Villous thrombus (patient numbers)	− (3)	++ (2) +++ (3) − (2)	+ (1) − (1)	+ (1) ++ (4) − (2)

+ mild degree; ++ moderate degree; +++ intense degree; − not present.

**Table 3 biology-12-00174-t003:** CD34 and PDPN immunohistochemistry reactivity in normal human placentas and SARS-CoV-2 placentas.

	Villous Stroma				Decidua Stroma				Endothelium			
	Group 1	Group 2	Group 3	Group 4	Group 1	Group 2	Group 3	Group 4	Group 1	Group 2	Group 3	Group 4
CD34	−	−	−	−	−	−	−	−	+++	+++	+++	+++
PDPN	++	++	++	±	++	++	±	+	−	−	−	−

−, no reactivity; ±, very weak reactivity; +, weak reactivity; ++, moderate reactivity; +++, strong reactivity; group 1, control; group 2, first trimester infection; group 3, second trimester infection; group 4, third trimester infection.

## Data Availability

Not applicable.

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
