# Peer review of "SARS-CoV-2 Infection: A Clinical and Histopathological Study in Pregnancy"

_biology, 2023, doi:10.3390/biology12020174_

Round 1
Reviewer 1 Report
Review of the manuscript titled “SARS-CoV-2 infection: a clinical and histopathological study in pregnancy”
Abstract: well written.
Keywords: please, remove the “multiple symptomatic treatments” that is too long as keyword and replace it with others.
Introduction: very well written and almost comprehensive. I think that this section could be improved still to add the following informations of these papers:
1) Resta L, Vimercati A, Cazzato G, Mazzia G, Cicinelli E, Colagrande A, Fanelli M, Scarcella SV, Ceci O, Rossi R. SARS-CoV-2 and Placenta: New Insights and Perspectives. Viruses. 2021 Apr 21;13(5):723. doi: 10.3390/v13050723. PMID: 33919284; PMCID: PMC8143362.
2) Jamieson DJ, Rasmussen SA. An update on COVID-19 and pregnancy. Am J Obstet Gynecol. 2022 Feb;226(2):177-186. doi: 10.1016/j.ajog.2021.08.054. Epub 2021 Sep 14. PMID: 34534497; PMCID: PMC8438995.
3) Vimercati A, Cazzato G, Fanelli M, Scarcella SV, Ingravallo G, Colagrande A, Sablone S, Stolfa M, Arezzo F, Lettini T, Rossi R. SARS-CoV-2, Placental Histopathology, Gravity of Infection and Immunopathology: Is There an Association? Viruses. 2022 Jun 18;14(6):1330. doi: 10.3390/v14061330. PMID: 35746801; PMCID: PMC9227044.
Material and methods: I have a couple of questions for the authors: firstly, why were pregnancies with fetal malformations or other problems excluded? Excluding maternal pathologies, are we sure this is not a selection bias? In my opinion, in the current state of knowledge, we do not have absolute proof that SARS-CoV-2 is not able to determine stillbirths, IUGR, etc. On the contrary, some works consider that positivity to the virus could represent a risk factor for these issues. Second problem: the number of placentas analyzed is too small. In some groups (such as - Infection during the second trimester of gestation, there are only 2 placentas.
Why not use the Amsterdam classification for the categorization of histopathological abnormalities of the placentas? And how were the forms of COVID-19 disease categorized in pregnant women? Has a standardized clinical scheme been followed which allows for the division of severity/symptoms into mild/moderate or severe?ntas!). This is a serious problem for possible generalization of the obtained results.
Table 1. Please, add a legend to help the readers to understand the significance of + symbols.
Line 167: “by nasopharyngeal swab”. What’s swab? PCR? Please, details!
Author Response
Dear Reviewer,
Thank you for your revision work on the manuscript.
Taking all comments into account, I made the requested corrections.
In Keywords
I removed the “multiple symptomatic treatments”.
In Introduction:
I improved the section by adding information from the following works:
- Resta L, Vimercati A, Cazzato G, Mazzia G, Cicinelli E, Colagrande A, Fanelli M, Scarcella SV, Ceci O, Rossi R. SARS-CoV-2 and Placenta: New insights and perspectives. Virus. 2021 Apr 21;13(5):723. doi: 10.3390/v13050723. PMID: 33919284; PMCID: PMC8143362.
- Jamieson DJ, Rasmussen SA. An update on COVID-19 and pregnancy. Am J Obstet Gynecol. 2022 Feb;226(2):177-186. doi: 10.1016/j.ajog.2021.08.054. Published 14 September 2021. PMID: 34534497; PMCID: PMC8438995.
- Vimercati A, Cazzato G, Fanelli M, Scarcella SV, Ingravallo G, Colagrande A, Sablone S, Stolfa M, Arezzo F, Lettini T, Rossi R. SARS-CoV-2, placental histopathology, severity of infection and immunopathology: Is there an association? Virus. 2022 Jun 18;14(6):1330. doi: 10.3390/v14061330. PMID: 35746801; PMCID: PMC9227044.
In material and methods:
1. Why were pregnancies with fetal malformations or other problems excluded?
We excluded pregnancies with fetal malformations or other problems, as our idea was to try to evaluate a potential effect of the infection in relation to normality, without having a 'contamination' of the results in relation to other pathologies.
2. Excluding maternal pathologies, are we sure this is not a selection bias?
Certainly, your opinion is correct and certainly the SARS-CoV-2 infection can induce alterations that can lead to stillbirths, etc. In our study, however, we preferred to make this type of selection to try to have as homogeneous a pool as possible for now, also because there are several conditions that could lead to serious alterations. The study will certainly continue, both to increase the number of cases and to investigate the mechanisms and thus also evaluate more serious cases.
3. The number of placentas analyzed is too small.
Yes, it is true that in group 3 the number of cases is very small. Unfortunately, when we selected the patients according to our criteria, only 2 cases fell into that group and we still preferred to include them. We are working on amplifying the number of cases. We have included in the discussion section this limitation to our work.
4. Why not use the Amsterdam classification for the categorization of histopathological abnormalities of the placentas?
Classical histopathological parameters for the classification of histopathological abnormalities were used for the analysis of placentas.
5. And how were the forms of COVID-19 disease categorized in pregnant women?
Has a standardized clinical scheme been followed which allows for the division of severity/symptoms into mild/moderate or severe?
To classify the severity of COVID-19 disease in pregnant women, different scientific societies proposed standardized clinical schemes.
We followed the American College of Obstetricians and Gynecologists (ACOG) classification, which assesses the severity/symptoms into mild/moderate or severe based on patient history, physical exam findings, laboratory tests, imaging studies, and patient response to treatment.
6. Please, add a legend to help the readers to understand the significance of + symbols.
A descriptive legend has been added for table 2.
7. Line 167: “by nasopharyngeal swab”. What’s swab? PCR? Please, details!
A description of the nasopharyngeal swab was added to line 167.
Reviewer 2 Report
Perna et al reported a cohort study to placenta pathology resulted from SARS-CoV-2 infection during pregnancy, including expression of vasculogenesis markers, fibrin deposits and lymphocyte infiltration. Importantly, samples were taken from individuals who got infected in different trimesters, and all three trimesters were covered, allowing a comprehensive analysis of impact on placental health by SARS-CoV-2 infection. However, the results of this manuscript are incomplete and may not be sufficient for publication at this point. Following are major concerns.
1. The secondary outcomes, as stated in the abstract, as well as the association between maternal symptoms/treatment and placental pathology were not mentioned in the results section. The quantification of immunohistochemistry, or HSCORE was not reported either. These should be included.
2. The sample size is small, specifically in Group 3 (n = 2). The author may elaborate potential impact on effect size, statistical power, etc.
3. The sampling method for placenta and replicates in immunohistochemistry should be included in methods or figure legends. For example, which part of placenta was retrieved? How sampling bias was minimized?
4. Please make sure all citations are proper. For example, citation Seelam et al [22] in line 159, page 5 is not the one in reference list. Please also finish the author contributions section.
Author Response
Dear Reviewer,
Thank you for your revision work on the manuscript.
Taking all comments into account, I made the requested corrections.
1. The secondary outcomes, as stated in the abstract, as well as the association between maternal symptoms/treatment and placental pathology were not mentioned in the results section. The quantification of immunohistochemistry, or HSCORE was not reported either. These should be included.
In the results section we added a descriptive section between maternal symptoms/treatment and placental pathology.
For immunohistochemistry, quantification was done, as described in material and methods.
2. The sample size is small, specifically in Group 3 (n = 2). The author may elaborate potential impact on effect size, statistical power, etc.
Yes, it is true that in group 3 the number of cases is very small. Unfortunately, when we selected the patients according to our criteria, only 2 cases fell into that group and we still preferred to include them. The study, for now, is mainly observational, but we are working on amplifying the number of cases and then carrying out a statistical study. We have included in the discussion section this limitation to our work.
3. The sampling method for placenta and replicates in immunohistochemistry should be included in methods or figure legends. For example, which part of placenta was retrieved? How sampling bias was minimized?
Placental sampling was performed following standard protocols and in a methodical way at numerous regions, such as chorionic plate, chorionic villi, basal plate, amniotic membrane, and umbilical cord (Venceslau EM et al, 2020).
4. Please make sure all citations are proper. For example, citation Seelam et al [22] in line 159, page 5 is not the one in reference list. Please also finish the author contributions section.
All citations were checked.
Authors' contributions section completed.
Round 2
Reviewer 1 Report
I would like to thank the academic publisher for giving me the opportunity to re-evaluate this manuscript I first saw. I believe that despite the limitations that have been elucidated by the authors, the manuscript can be accepted.
Reviewer 2 Report
All my points are well responded in the revised edition, And I'm happy to recommend acceptance of current manuscript.